# Learning from the past: A short term forecast method for the COVID-19 incidence curve

**Jean-David Morel**[1]☯, **Jean-Michel Morel** ⬵[2]☯*, **Luis Alvarez**[3]☯

**1** Laboratory of Integrative Systems Physiology, Ecole Polytechnique Fédérale de Lausanne, EPFL/IBI/LISP Lausanne, Switzerland, **2** ENS Paris-Saclay, CNRS, Centre Borelli, Université Paris-Saclay, Gif-sur-Yvette, France, **3** Departamento de Informática y Sistemas, Universidad de Las Palmas de G.C., Las Palmas de G. C., Spain

☯ These authors contributed equally to this work.
* jean-michel.morel@ens-paris-saclay.fr

## Abstract

The COVID-19 pandemy has created a radically new situation where most countries provide raw measurements of their daily incidence and disclose them in real time. This enables new machine learning forecast strategies where the prediction might no longer be based just on the past values of the current incidence curve, but could take advantage of observations in many countries. We present such a simple global machine learning procedure using all past daily incidence trend curves. Each of the 27,418 COVID-19 incidence trend curves in our database contains the values of 56 consecutive days extracted from observed incidence curves across 61 world regions and countries. Given a current incidence trend curve observed over the past four weeks, its forecast in the next four weeks is computed by matching it with the first four weeks of all samples, and ranking them by their similarity to the query curve. Then the 28 days forecast is obtained by a statistical estimation combining the values of the 28 last observed days in those similar samples. Using comparison performed by the European Covid-19 Forecast Hub with the current state of the art forecast methods, we verify that the proposed global learning method, *EpiLearn*, compares favorably to methods forecasting from a single past curve.

## Author summary

Forecasting the short time evolution of the COVID-19 daily incidence is a key issue in the epidemic decision making policy. We propose a machine learning method which forecasts the future values of the daily incidence trend based on the evolution of other incidence trend curves that were similar to the current one in the past. Using comparison performed by the European Covid-19 Forecast Hub with the current state of the art forecast methods, we verify that the proposed global learning method, *EpiLearn* compares favorably to methods that forecast from a single past curve.

**Data Availability Statement:** EpiLearn, the forecasting model presented in this work, is implemented in the publicly available EpiInvert CRAN R package https://cran.r-project.org/web/packages/EpiInvert/index.html In this package,

EpiLearn is executed using the EpiInvertForecast R function. A description of EpiInvertForecast usage is presented with examples in the vignette https://ctim.ulpgc.es/covid19/EpiInvertForecast.html The incidence trend database has been built using the daily incidence data, up to May 5, 2022, provided by Our World in Data https://ourworldindata.org/coronavirus for the following countries and regions: Argentina, Austria, Bangladesh, Belgium, Brazil, Canada, Chile, Colombia, Cuba, Czech Republic, Denmark, Germany, France, Greece, Hungary, India, Iraq, Iran, Ireland, Israel, Italy, Japan, Jordan, Kazakhstan, Malaysia, Mexico, Nepal, Netherlands, Peru, Philippines, Poland, Romania, Russia, Serbia, Slovakia, South Africa, South Korea, Spain, Sweden, Switzerland, Thailand, Tunisia, Turkey, Ukraine, United Arab Emirates, United Kingdom, USA, Vietnam, Africa, South America, North America, Asia, Europe, European Union, Oceania, and the world. The database provided by Our World in Data includes COVID-19 information about confirmed cases, deaths, vaccinations, testing and government responses. Confirmed cases and deaths are collected by Johns Hopkins University by date of report, rather than date of test/death. The comparative results with other methods have been obtained by using the weekly evaluation reports published by the European Hub, in the repository https://github.com/covid19-forecast-hub-europe/covid19-forecast-hub-europe/tree/main/evaluation/weekly-summary This information is given in the material and methods section of the paper.

**Funding:** JMM was sponsored by Kayrros, Kayrros.com under Grant 2018-0143. The funders had no role in study design, data collection and analysis, decision to publish, or preparation of the manuscript.

**Competing interests:** The authors have declared that no competing interests exist.

This is a *PLOS Computational Biology* Methods paper.

## Introduction

The COVID-19 epidemic has provided us with information on the evolution of the daily incidence in many different countries and epidemic scenarios. Given the enormous global impact of COVID-19, a large number of researchers have studied the problem of predicting the incidence curve. For example, the European Covid-19 Forecast Hub [1] gathers a variety of prediction models based on many different techniques. These methods observe the past of daily incidence in a given country and forecast its future evolution in the weeks to come. The prediction is generally made for the next four weeks. Most of these methods base their forecast on the observation of only the past values of the current incidence curve, that is, the one that they want to extend towards the future.

The main objective of this paper is to introduce a prediction method that learns the future of a given incidence trend curve from the past evolution of other many incidence trend curves. Our method can be seen as an extension of the "method of analogues", inspired from meteorology and first introduced for epidemiologic forecasting in [2] in predicting influenza activity.

This method uses vectors selected from historical influenza time series that match current activity. The authors applied it to forecasting the incidences of influenza in France and in the country's 21 administrative regions, using a series of data for 938 consecutive weeks of surveillance between 1984 and 2002, and compared the results with those for autoregressive models. They reported that for 1- to 10-week-ahead predictions, the correlation coefficients between the observed and forecasted regional incidences was significantly superior with the method of analogues than for autoregressive models. The method compares fixed incidence intervals to a query interval by their Euclidean distance, and obtains a prediction as a weighted mean of the incidences that follow the nearest neighbors. Nevertheless, a major difference of their method with ours is that they restrict their comparison to the past history of each incidence curve. Hence, their learning set is considerably smaller than the one that uses many regions or countries: It assumes the observation of a several years period and takes advantage of the periodicity of influenza.

The sophisticated extension of the method of analogues proposed in [3] also uses historical data (up to 20 years) to obtain predictive distributions for incidence in individual weeks using a kernel conditional density estimation (KCDE). Then these individual distributions are tied into joint distributions using copulas, to predict the timing of and incidence in the peak week of the season. Like in [2], the method is applied to a single time series and therefore requires a much longer observation period that the one that could be used for COVID-19 so far. Arguably the closest method to our proposed one is the neural method of [4]. The authors introduce a new neural forecasting model called Attention Crossing Time Series, that makes forecasts via comparing patterns across time series obtained from multiple regions. It interprets the attention mechanism [5] as an application of the "method of analogues". The model is demonstrated to outperform many recent SEIR models.

In a nutshell, our proposed learning method uses all past incidence trend curves that are similar on 28 consecutive days to the last 28 last days of the trend incidence curve that is to be extended towards the future. To demonstrate the method, we use as learning database a collection of 27,418 COVID-19 past incidence trend curves across 61 world regions and countries. These trend curves are computed by the *EpiInvert* method [6] from the original raw incidence curves communicated by the governments. A raw incidence curve is not the adequate input

for forecasting because of its high noise and weekly oscillation. The weekly seasonality depends on each country, thus hindering comparison between raw incidence curves. Trend curves instead, being freed from seasonality and noise, are much more suitable to forecasting. Nevertheless, as we will show later, a daily forecast of the raw incidence can be deduced from its forecasted trend using the estimated seasonality.

Let us denote by $\mathbf{s} = (s_1, s_2, \ldots, s_{28})$, the last 28 values of the current incidence trend that we want to extrapolate, and by $\mathbf{s}^f = (s_1^f, s_2^f, ..., s_{28}^f)$ the forecast for the next 28 days proposed in this work. Each of the 27,418 incidence trend curves in our database contains the values of 56 consecutive days extracted from observed past incidence curves. We predict the evolution of the current incidence trend curve from the median of the 28 last days of the 27,418 database curves, where the median is computed on the 121 most similar curves. The similarity to the query of these candidate curves is measured on its first 28 days, which are matched to the 28 last observed days of the query curve $\mathbf{s}$ that we want to extrapolate. In summary, the 28 future samples $\mathbf{s}^f$ of the current curve $\mathbf{s}$ are obtained as the median of the corresponding days 29 to 56 of the most similar past curves. Alternatively, we also tested a weighted average of all curves instead of the median, but it has a slightly inferior performance.

We also compute empirical confidence intervals for the incidence trend forecast by applying the proposed method to the incidence curves of our database and obtaining a distribution of the forecast error as a function of the number of days passed from the current day (the last day of the used incidence curve). In Fig 1 we illustrate the results of the proposed method for four countries, using their incidence curves up to May 5, 2022. This figure displays in black the raw input incidence curves, which show a strong weekly periodic bias. In the case of France for example, there is a strong deficit on week-ends compensated by a peak on Mondays. For our prediction, we therefore use a smooth incidence trend curve (in red), that is easier to extend and forecast than the original raw incidence. The usual way to compute an incidence trend curve is to apply a 7 or 14 days sliding average to the original raw incidence, which reduces the weekly effects [7]. In our method, we use the more sophisticated *EpiInvert* method introduced in [6, 8] and available as a CRAN R package [9]. This method is summarily described in the Material and Methods section. Fig 1 shows in blue the forecast curve, that can be compared to the magenta ground truth that became later available. In light blue, the figure also displays the predicted raw incidence curve where the weekly bias learned by *EpiInvert* in the immediate past is also applied. In these relatively favorable examples, picked from large countries with large incidence and at a time of regular daily measurements, the error between ground truth and prediction seems acceptable. Nevertheless, the error on the fourth week can exceed 25%. This is not surprising, given the high variability of the possible futures depending on human interventions. In this introduction we do not present the many alternative forecasting methods. Instead, we review them in detail in the discussion section. The methods that were publicly available through the European Covid-19 Forecast Hub are quantitatively compared with our method through the unbiased metrics of the hub. Our learning technique is different in structure from most previous methods introduced in the literature. We involve no parametric model for the incidence curve. Our method produces a daily forecast of the future, whereas most COVID-19 incidence analysis methods [7, 10] aim to forecast the 7-day sliding average of the daily incidence.

The particular significance of this study lies in the novelty of our machine learning approach that provides a daily forecast of the current incidence curve based on its similarity with many different incidence curves in the past. The unbiased comparison with other methods in the context of the European Covid-19 Forecast Hub confirms the good performance of the proposed method.

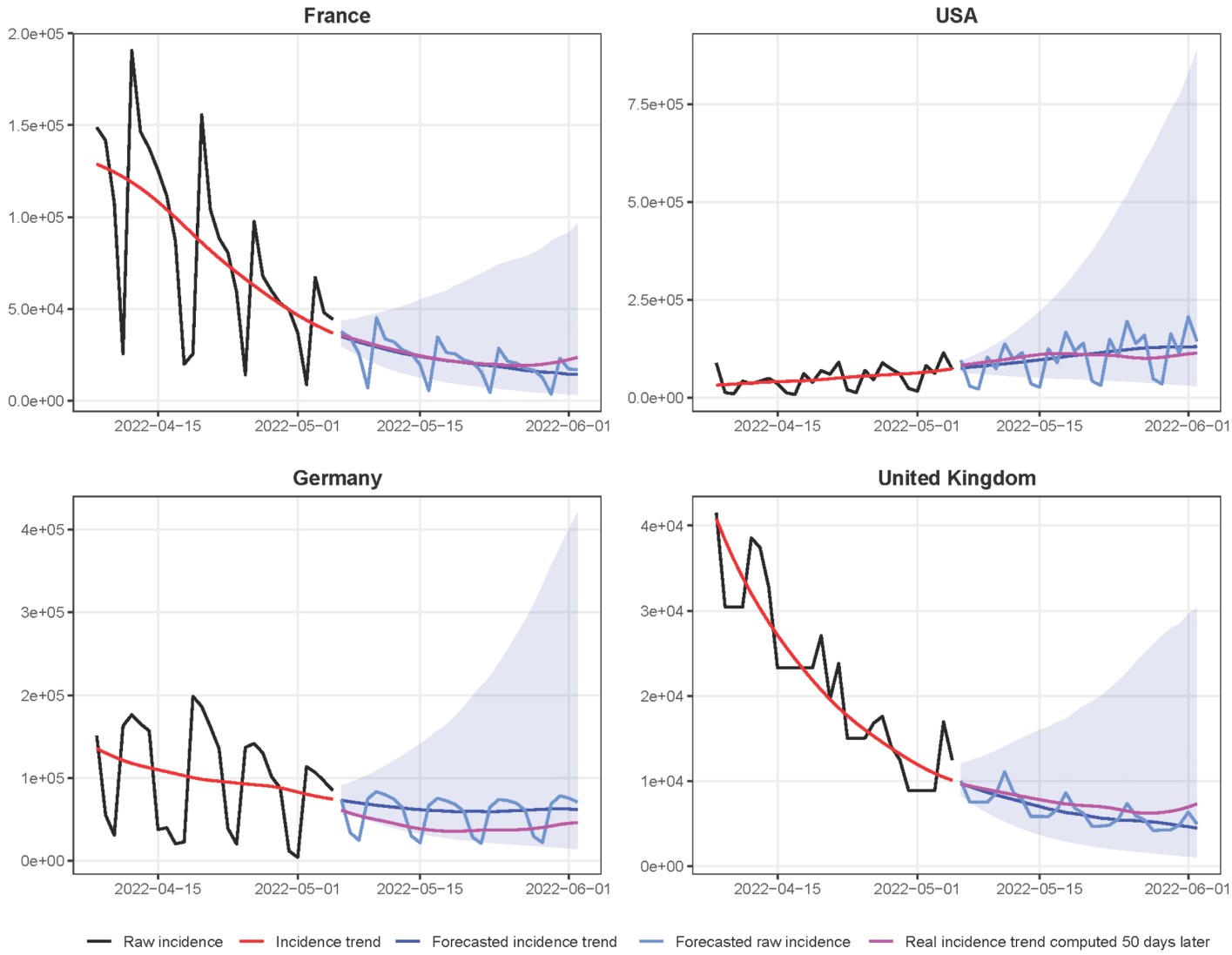

**Fig 1. 28-day forecast of the daily incidence for four countries, using the data up to May 5, 2022.** The current original raw incidence curve which suffers from periodic weekly effects. In **red** the current incidence trend computed by *EpiInvert* [6], in **blue** the forecast of the incidence trend curve by *EpiLearn*, in **magenta** the ground truth given by the incidence trend curve obtained 50 days later and in **light blue** the forecast of the raw incidence using Eq (8). The **cyan** shaded area represents a 95% empirical confidence interval of the incidence trend forecast. The discontinuity at the past-future junction in Germany is due to a sharp drop of the incidence after the last observed day. When recalculating the incidence trend curve, the values of the past days are also changed by smoothing, thus creating the observed gap.

## Material and methods

### Data sources

The incidence trend database has been built using the daily incidence data, up to May 5, 2022, provided by *Our World in Data* in [11] for the following countries and regions: Argentina, Austria, Bangladesh, Belgium, Brazil, Canada, Chile, Colombia, Cuba, Czech Republic, Denmark, Germany, France, Greece, Hungary, India, Iraq, Iran, Ireland, Israel, Italy, Japan, Jordan, Kazakhstan, Malaysia, Mexico, Nepal, Netherlands, Peru, Philippines, Poland, Romania, Russia, Serbia, Slovakia, South Africa, South Korea, Spain, Sweden, Switzerland, Thailand, Tunisia, Turkey, Ukraine, United Arab Emirates, United Kingdom, USA, Vietnam, Africa, South America, North America, Asia, Europe, European Union, Oceania, and the world.

The database provided by *Our World in Data* in [11] includes COVID-19 information about confirmed cases, deaths, vaccinations, testing and government responses. Confirmed cases and deaths are collected by Johns Hopkins University by date of report, rather than date of test/death. Therefore, the number they report on a given day does not necessarily represent the actual number on that date, because of the long reporting chain that exists between a new case/death and its inclusion in statistics. This also means that time series can show sudden changes (negative or positive) when a country corrects historical data, because it had previously under-or -over estimated the number of cases/deaths. The comparative results with other methods, presented in the comparative results part of the Results section, have been obtained by using the raw evaluation scores published by the European Hub, [1], in the file https://github.com/covid19-forecast-hub-europe/covid19-forecast-hub-europe/tree/main/evaluation/scores.csv. These results cannot be manipulated and they use the version of the data that have been available in real time when producing forecasts.

## Data preprocessing method

The infectiousness of individuals at time $t$ is characterized by the reproduction number $R_t$, defined as the average number of cases generated by an infected person at time $t$, and by the (observable) serial interval $\Phi_s$ which represents the time distribution of the delay of the onset of symptoms between primary and secondary cases. For Covid 19, this serial interval was measured accurately in [12] on nearly 1000 verified transmission pairs. We use this distribution in the EpiInvert method.

Our forecast model uses the EpiInvert method, which aim it is to invert the fundamental renewal equation [13, 14] linking $R_t$, $\Phi$ and the incidence $i_t$ of new daily cases,

$$i_t = \sum_s i_{t-s} R_{t-s} \Phi_s \quad \text{for} \quad t = 0, .., t_c, \tag{1}$$

where $t_c$ is the current time. The EpiInvert method introduced in [8] and extended in [6, 9] is a deconvolution + denoising procedure to solve the functional Eq (1). EpiInvert estimates both $R_t$ and a restored $i_t$ corrected for the weekend bias. To remove the weekend effect, it computes a 7-day quasi-periodic multiplicative factor $q_t$. From the observed incidence curve and the serial interval, $R_t$ and $q_t$ are jointly estimated by minimizing

$$E(R, q) = \sum_{t=0}^{t_c} \left( \frac{q_t i_t^0 - \sum_s q_{t-s} i_{t-s}^0 R_{t-s} \Phi_s}{median_{(t-\tau, t]}(i^0)} \right)^2 + w_R \sum_{t=1}^{t_c} (R_t - R_{t-1})^2 + w_q \sum_{t=7}^{t_c} (q_t - q_{t-7})^2, \tag{2}$$

where $median_{(t-\tau, t]}(i^0)$ is the median of $i_t^0$ in the interval $(t - \tau, t]$ used to normalize the energy with respect to the size of $i_t$ (the value of $\tau$ is fixed to 21 (3 weeks) in the experiments). The total number of cases is preserved by adding to (2) the constraint on $q_t$:

$$\sum_{t=t_c-T+1}^{t_c} i_t^0 = \sum_{t=t_c-T+1}^{t_c} q_t i_t^0, \tag{3}$$

where $T$ is a period of analysis empirically fixed to $T = 56$ days. The minimization of the above energy yields estimates of $R_t$, and a quasi-periodic bias correction factor $q_t$, as the third term in the functional forces the values $q_t - q_{t-7}$ to be small. The parameters $w_R$ and $w_q$ are regularization weights with default values $w_R = w_q = 5$. Their values were proven in [8] to be nearly optimal for Covid-19 incidence curves.

By minimizing this energy (2) under the constraint (3), we obtain the reproduction number $R_t$ and the seasonality bias correction coefficients $q_t$. An incidence $\hat{i}_t$ corrected of the weekly

bias is obtained as $\hat{i}_t = q_t i_t^0$. The final restored incidence $i_t$ that we use for forecasting is obtained by applying the renewal Eq (1) to the bias corrected incidence $q_t i_t^0$, namely

$$i_t =: \sum_s q_{t-s} i_{t-s}^0 R_{t-s} \Phi_s.$$

## Forecasting method

Next, we present the proposed method, let us, first, to introduce the following notation to manage the incidence curves and their forecast:

- $\mathbf{s}^o = (s_1^o, \ldots, s_{28}^o)$ : the current raw incidence curve in the last 28 days.

- $\mathbf{s}^{o,f} = (s_1^{o,f}, \ldots, s_{28}^{o,f})$ : the forecast of the current raw incidence curve for the next 28 days in the future.

- $\mathbf{s} = (s_1, \ldots, s_{28})$ : the current trend curve in the last 28 days.

- $\mathbf{s}^f = (s_1^f, \ldots, s_{28}^f)$ : the forecast of the incidence trend curve for the next 28 days in the future.

- $\{\mathbf{i^k} = (i_1^k, \ldots, i_{56}^k)\}_{k=1}^N$ : the collection of incidence trend curves in the database.

- $\{\mathbf{s}^k = (i_1^k, \ldots, i_{28}^k)\}_{k=1}^N$ : the first 28 days of the database incidence trend curves that we use for comparison purposes with the current curve $\mathbf{s}$.

- $\{\mathbf{s}^{f,k} = (s_1^{f,k}, \ldots, s_{28}^{f,k})\}_{k=1}^N$ : the forecast of the database curves for the last 28 days using as current curve the first 28 days.

- $E_d = \{e_{d,k} = \frac{|s_d^{f,k} - i_{d+28}^k|}{s_d^{f,k}}\}_{k=1}^N$ : empirical distribution of the relative forecast error for the database curves in the forecast day $d = 1, \ldots, 28$.

## Incidence trend curves database construction using *EpiInvert*

Our proposed method, *EpiLearn*, uses a world-wide database of raw incidence curves from 61 countries and regions up to May 5, 2022. For each country or region, and for each day, starting 150 days after the beginning of the epidemic, we take the raw incidence data up to that day. Then, the resulting curve is further processed by applying the *EpiInvert* incidence decomposition algorithm [6] (see the Material and methods section) and we keep the last 56 values of the estimated incidence trend curve. To add a curve of this type to the database, we impose that the mean of the 56 values of the sequence must be larger than 1000. Taking into account that we normalize all database curves, the magnitude of the curves therefore has no influence in the forecast estimation. This amounts to making the assumption that the incidence curve evolution has the same behavior in large countries than in small countries. We impose this minimum 1000 cases average condition because for very small averages the registered incidence curves are often very noisy and unreliable. Indeed, small averages often correspond to non-threatening or neglected stages of the epidemic.

## Normalization of the database incidence curves

*EpiInvert* is magnitude-invariant, that is, multiplying the raw incidence values by a scalar factor multiplies the estimated EpinInvert incidence trend values by the same scalar factor. Our

forecast method preserves this magnitude-invariance by normalizing the magnitude of the incidence trend curves.

Let $N$ be the number of incidence trend curves stored in the database (in our case $N = 27{,}418$). For $k = 1, 2, .., N$, $\mathbf{i}^k = (i_1^k, i_2^k, ..., i_{56}^k)$ corresponds to the last 56 days of the incidence trend curve computed by *EpiInvert* and stored in the database. Each $\mathbf{i}^k$ has been normalized by multiplying it by a scale factor so that the average of the first 28 values be equal to 1:

$$\frac{\sum_{j=1}^{28} i_j^k}{28} = 1. \tag{4}$$

**Computing the distance between curves.** We denote by $\hat{\mathbf{s}}$ the present-day incidence trend curve for the country being predicted, that has been normalized in the same way, so that

$$\hat{\mathbf{s}} = \frac{28}{\sum_{j=1}^{28} s_j} \mathbf{s}. \tag{5}$$

We compare the normalized vectors $\hat{\mathbf{s}}$ and $\mathbf{s}^k$ (the first 28 values of $\mathbf{i}^k$) through the following magnitude-invariant distance average,

$$d(\hat{\mathbf{s}}, \mathbf{s}^k, \mu) = \frac{\sum_{j=1}^{28} e^{-\mu(28-j)} |\hat{s}_j - s_j^k|}{28}, \tag{6}$$

where the parameter $\mu \geq 0$ governs the exponentially weighted moving average. The larger the value of $\mu$, the lower this weight for the more remote days, as is classical in control theory [15] and in epidemiological forecasting [16]. As shown below, by minimizing the forecast error, we obtain the optimal value $\mu = 0.0475$. Fig 2 shows the function $f(x) = e^{-0.0475x}$ which determines the weight assigned to each day in the past in the distance estimation.

## Forecasting using a median of the closest database curves

First, we select in the database the $N_{median}$, the curves $\{\mathbf{s}^{k_n}\}_{n=1,..,N_{median}}$ that are closest to the current one, using the similarity criterion (6). $N_{median}$ is a parameter of the method. The median

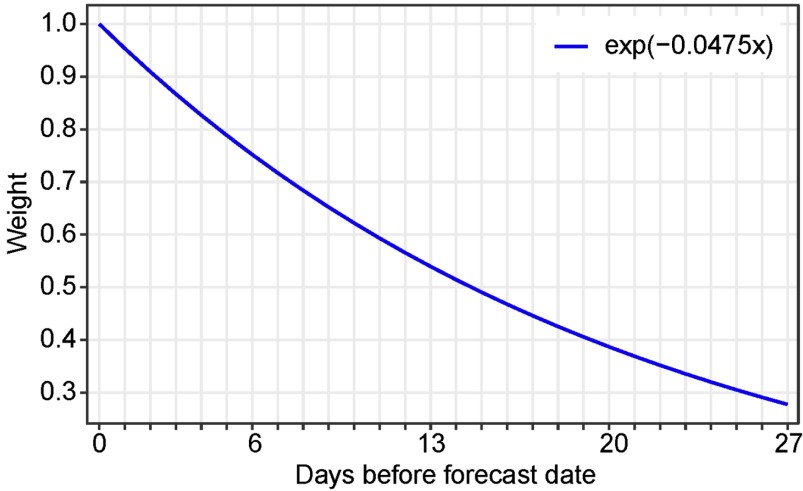

**Fig 2. Weighting function.** Shape of the functions $e^{-0.0475x}$ which determines the weight assigned to each day in the past in the distance estimation (6) for the proposed forecast method.

forecast of $\mathbf{s}^f = (s_1^f, s_2^f, ..., s_{28}^f)$ for the next 28 days is defined by

$$s_d^f = median\left\{\frac{s_{28}}{s_{28}^{k_n}} i_{d+28}^{k_n}\right\}_{n=1,...,N_{median}} \qquad \text{for } d = 1, .., 28. \tag{7}$$

As *EpiInvert* also computes multiplicative weekly seasonality correction factors, $q_t$, we additionally compute a forecast, $\mathbf{s}^{o,f}$, of the raw incidence curve, $\mathbf{s}^o$, by dividing the forecasted incidence trend curve by the corresponding seasonality factors,

$$s_d^{o,f} = \frac{s_d^f}{q_{22+d\%7}} \qquad \text{for } d = 1, .., 28. \tag{8}$$

where % is the modulus operator. By using $q_{22+d\%7}$ as future seasonality factors we are simply making a 7-periodic extrapolation of the last seasonality factors estimated by *EpiInvert*.

Fig 3 illustrates the proposed learning procedure. For four countries, it shows the current incidence trend by EpinInvert, its 5 closest curves in the database for their first four weeks, and the forecast, computed as the median of the 121 closest curves in their last four weeks. For France, the UK and the USA, we can observe that among the most similar curves there are curves with a strong growth. These curves correspond to the first wave of the omicron variant in Romania, Hungary and Italy that occurred by the end of 2021. These examples show that very close curves in the past can evolve very differently in the future. In particular, the methods studied in this paper, which forecast the evolution of the incidence only using past incidence data, may be subject to large errors in forecasting.

## Choice of the method parameters

We have to choose the parameters of the method, $N_{median}$ and $\mu$. For each curve $\mathbf{i}^k = (i_1^k, \ldots, i_{56}^k)$ in the database, we use as current incidence curve the first 28 days of $\mathbf{i}^k$, that is, $\mathbf{s}^k = (i_1^k, \ldots, i_{28}^k)$. The forecast of $\mathbf{s}^k$ is given by

$$s_d^{f,k} = median\left\{\frac{s_{28}^k}{s_{28}^{k_n}} i_{d+28}^{k_n}\right\}_{n=1,...,N_{median}} \qquad \text{for } d = 1, .., 28, \tag{9}$$

where $\{\mathbf{s}^{k_n}\}_{n=1}^{N_{median}}$ are the $N_{median}$ closest curves, in the database, using the distance (6), to $\mathbf{s}^k$ (removing from the choice a neighborhood of $k$ in the database). For each forecast day $d = 1, .., 28$, the relative forecast error is given by

$$e_{d,k} = \frac{|s_d^{f,k} - i_{d+28}^k|}{s_d^{f,k}}. \tag{10}$$

We define the method's median forecast error by

$$\text{ForecastError}(N_{median}, \mu) = median\left\{\frac{1}{14}\sum_{d=1}^{14} e_{d,k}\right\}_{k=1,..,N}. \tag{11}$$

By minimizing this median error, we obtained the optimal values $N_{median} = 121$ and $\mu = 0.0475$. We optimized the parameters $N_{median}$ and $\mu$ using the first 14 forecast days because the expected error in the next 14 days is so large that we prefer to focus on the optimization for the first 14 days. We could also optimize the above parameters for the whole 28 forecast days. In that case, we obtain as optimal values $N_{median} = 128$ and $\mu = 0.1075$ which are slightly different from the ones obtained for the first 14 days.

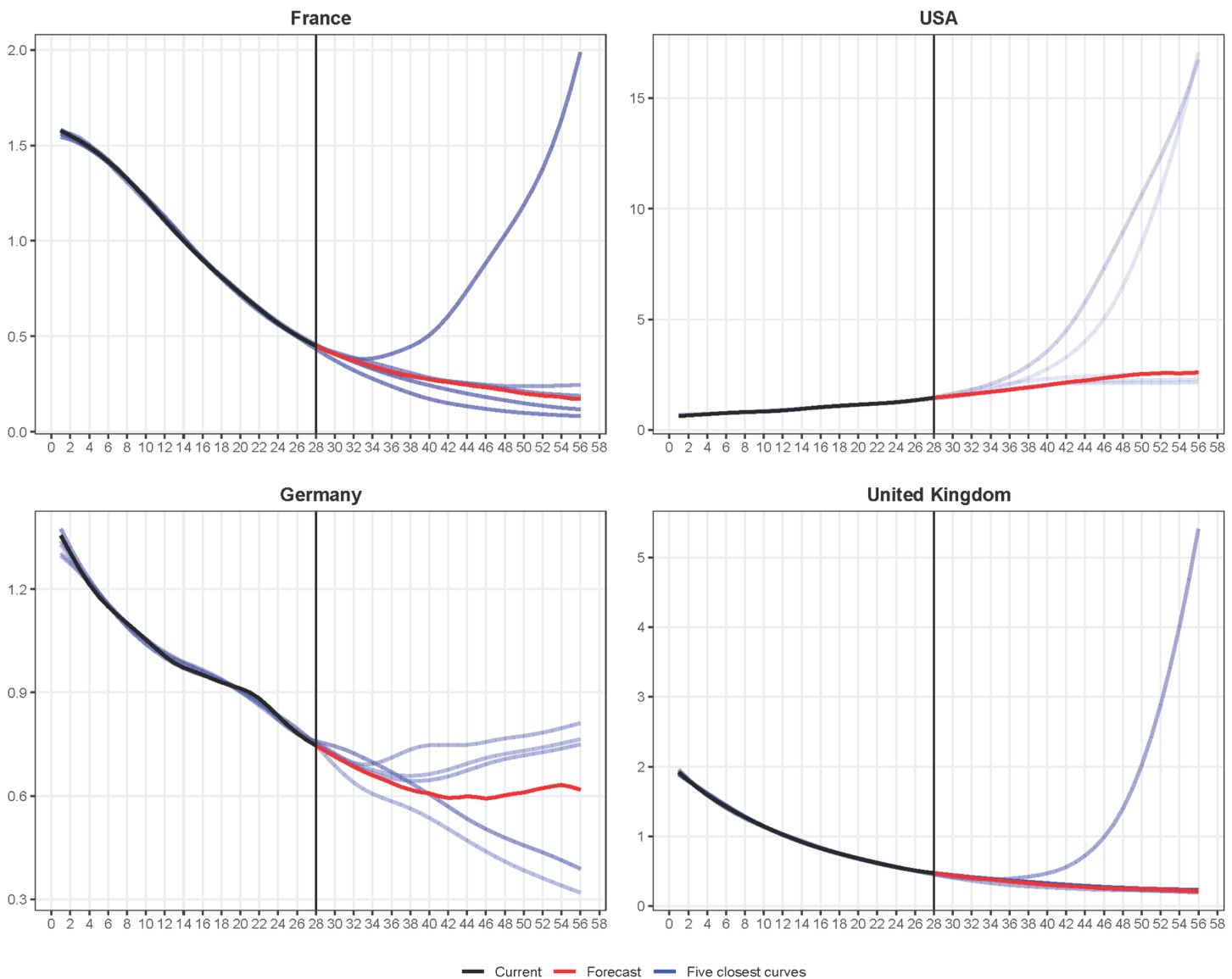

**Fig 3. Illustration of the variability of closest curves.** For France, the USA, Germany and the United Kingdom: in **black**, the normalized curve ŝ of the last 28 values of the incidence trend curve up to May 5, 2022, in red the normalized forecasting curve û obtained by *EpiLearn*. Are also displayed in a blue scale the five curves $\mathbf{i}^k$ in the database with the lowest distance $d(\hat{\mathbf{s}}, \mathbf{i}^k, \mu)$ to the incidence trend curve ŝ. The lighter the blue, the larger the distance to the input curve.

### Empirical confidence intervals

For each forecast day $d$, we compute empirical confidence intervals using the distribution of relative errors for the database given by $E_d = \{e_{d,k}\}_{k=1}^N$, using the estimated optimal values for the parameters $N_{median}$ and $\mu$.

Assuming that the distribution of the relative forecast error for the current incidence trend curve **s** is similar to the one obtained for the database and determined by $E_d$, we can empirically approximate the percentiles of the forecast distribution, $F_d$, of the current curve, using the percentiles of $E_d$. Indeed, let us denote by $P_p(X)$ the $p$-th percentile of a distribution $X$, then

$$P_p(F_d) \approx s_d^f + s_d^f P_p(E_d) \tag{12}$$

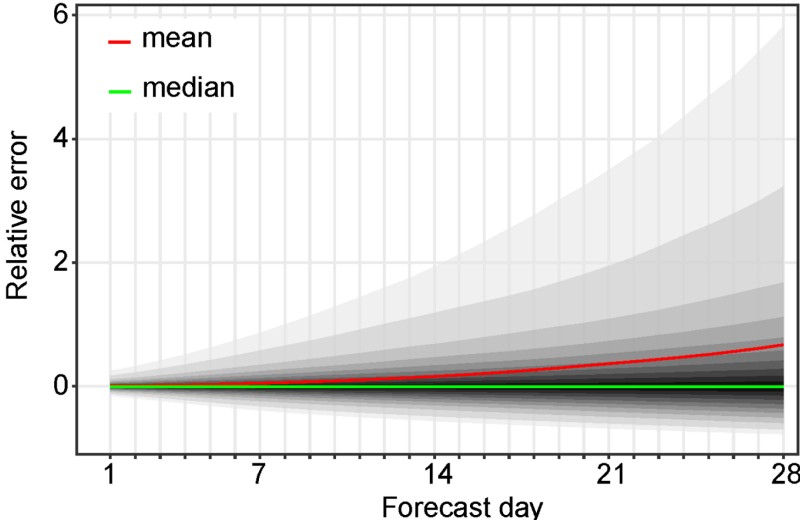

**Fig 4. Error statistics.** Illustration of some statistics of the $E_d = \{e_{d,k}\}$ distribution defined by (10) for the entire database: the **red** curve indicates the mean of the distribution that is greatly affected by the skewness of the distribution, which justifies using the median (the curve in **green**) instead of the mean. The median is indeed very close to zero, which proves the consistency of the approximation adopted in Eq (12). From the outside to the inside, the shaded areas represent the estimated $(1 - \alpha_k) \times 100\%$ central prediction intervals $(l_{\alpha_k}, u_{\alpha_k})$ for $\alpha_k = 0.05, 0.1, 0.2, \ldots,$ 0.9.

where $s_d^f$ is the forecast estimated by the proposed method. A 95% central confidence interval for the incidence trend value is for example given by $(P_{0.025}(F_d), P_{0.975}(F_d))$. In Fig 4 we display the confidence intervals of $E_d$ for the proposed forecast method. As expected, the size of the confidence intervals increases with the forecast day $d$ and is quite large after 28 days. Notice that the mean and the median $(P_{0.50}(E_d))$ of $E_d$ are very different due to the asymmetry of the distribution $E_d$. The mean is closer to the upper end of the forecast interval than the median. The fact that the median of the error is very close to zero confirms the consistency of the method.

## Results

### Comparative results in the context of the European Covid-19 Forecast Hub

The question arises of how to compare all methods, in theory and in practice. For a practical comparison, we take advantage of the fact that a wide variety of forecasts are submitted to the European COVID-19 Forecast Hub [17] and to the COVID-19 Forecast Hub [18]. A study on the methodology to evaluate and compare forecast has been proposed in [19], using the data of this Hub. As developed in [1], the European Covid-19 Forecast Hub provides short-term forecasts of Covid-19 cases and deaths across Europe. It is supported by teams working on pandemic modeling and sharing their forecast of the weekly accumulated incidence with horizons of 1 to 4 weeks. Each week starts on Sunday and ends on Saturday. At the time of writing, many countries do not provide data during the week-end, and some countries only provide a weekly estimate. This fact has no influence for method preprocessing the data by a 7 day sliding average. Nevertheless, since we use daily estimates, a single weekly estimate has a negative impact on the quality of our forecast. To address this issue, when a country provides data on a day, but not on the previous days, we distribute equally the last accumulated value over the previous uninformed days before applying *EpiInvert*.

Since *EpiLearn* forecasts the daily incidence, the weekly forecast is obtained by summing the forecasted raw daily incidence given by (8). The quantiles of the associated weekly distributions are computed on the registered database of incidence curves by extending the procedure of the previous section which computes the confidence intervals of the forecasted incidence curve. In this case, we aggregate to the weekly scale first and then compute quantiles.

The European Hub encourages teams to provide, for each model, $m$, each horizon week, $h = 1, 2, 3, 4$, and each forecast target, $n$, the prediction of the weekly incidence, $f_{m,h,\,n}$, and 23 quantiles of the associated distribution. These quantiles correspond to the predictive median, $M$, and eleven $(1 - \alpha_k) \times 100\%$ central prediction intervals $(l_{\alpha_k}, u_{\alpha_k})$, with $\alpha_k = 0.02, 0.05, 0.1, 0.2, \ldots, 0.9$, where $l_{\alpha_k}$ and $u_{\alpha_k}$ are (respectively) the $\alpha_k/2$ and $(1 - \alpha_k/2)$ quantiles of $F$. The following weighted interval score, $WIS_{m,h,n}$ (see [20]), is proposed to evaluate the distribution accuracy:

$$WIS_{m,h,n} = \frac{\frac{1}{2}|o_{h,n} - M| + \sum_{k=1}^{11} \frac{\alpha_k}{2}(u_{\alpha_k} - l_{\alpha_k}) + (l_{\alpha_k} - o_{h,n})_+ + (o_{h,n} - u_{\alpha_k})_+}{11.5} \quad (13)$$

where $o_{h,n}$ is the observed outcome, $(.)_+$ is defined as $(x)_+ = x$ if $x > 0$ and 0 otherwise. The lower the value of $WIS_{m,h,n}$, the better the score associated to the forecast distribution determined by the quantiles of $F$.

The prediction accuracy of a model is measured using two indicators: the first one is $|f_{m,h,n} - o_{h,n}|$, that is, the absolute value of the difference between the observed value $o_{h,n}$ and the prediction $f_{m,h,n}$. The second indicator measures the quality of the confidence intervals and is given by $WIS_{m,h,n}$. To compare the prediction accuracy of different models, we have to take into account that, in general, each team provides a different number of forecast targets. We started for example submitting forecast to the European Hub by August 2022, but other teams started submitting up to 2 years earlier. Furthermore, not all teams provide a forecast for all horizons and for all countries. Thus, defining a fair comparison of models requires some caution. To address this issue, the European Hub uses the following procedure (we explain the procedure for the comparison of $|f_{m,h,n} - o_{h,n}|$, but the comparison of $WIS_{m,h,n}$ is equivalent). Consider two models $m$ and $m'$, a week horizon $h \in \{1, 2, 3, 4\}$ and $\{(f_{m,h,n}, f_{m',h,n}, o_{h,n})\}_{k=1}^{N_{m,m',h}}$ where $N_{m,m',h}$ is the number of forecast targets that have been handled by both models. The pairwise comparison of both models is then defined by the ratio

$$\theta_{m,m',h} = \frac{\sum_{k=1}^{N_{m,m',h}} |f_{m,h,n} - o_{h,n}|}{\sum_{k=1}^{N_{m,m',h}} |f_{m',h,n} - o_{h,n}|}, \quad (14)$$

which is smaller than 1 if $m'$ is more accurate than $m$, and larger than 1 otherwise. Subsequently, we compute for each model $m$ the geometric mean of the results achieved for all different pairwise comparisons,

$$\theta_{m,h} = \left(\prod_{m'=1}^{M'} \theta_{m,m',h}\right)^{\frac{1}{M'}}, \quad (15)$$

where $M'$ is the number of models, $m' \neq m$, which have forecast targets in common with model $m$. It follows that $\theta_{m,h}$ is a measure of the relative skill of model $m$ with respect to the set of all other models in the week horizon $h$. The relative performance of model $m$ is

computed with respect to $\theta_{b,h}$, the score of the baseline model, as

$$\theta^*_{m,h} = \frac{\theta_{m,h}}{\theta_{b,h}}, \tag{16}$$

where the baseline model $b$ is nothing but the constant prediction extending the last observed weekly value [21].

The ratio $\theta^*_{m,h}$ is called the relative MAE, rel_ae, of model $m$ in the week horizon $h$. A score of $0 < rel\_ae < 1$ means that model $m$ is better than the baseline; a score of $rel\_ae > 1$ means that the baseline is better. In the case of $WIS_{m,h,n}$, we use the same procedure (replacing $|f_{m,h,n} - o_{h,n}|$ by $WIS_{m,h,n}$) and call rel_wis the associated indicator. Every week, the Hub publishes, in the file *scores.csv* of the evaluation repository, information about the accuracy of the predictions. In particular, it publishes, for each team $m$, horizon $h$ and forecast target $n$, the values of $|f_{m,h,n} - o_{h,n}|$, $WIS_{m,h,n}$, and the 50% and 95% prediction intervals coverage. We used this information to compare *EpiLearn* with the other methods. To do a fair comparison, for all teams, we used as comparison population the horizons and targets used by *EpiLearn* to provide forecast between August 6, 2022 and March 6, 2023. In this way we used for all teams the same target population when computing the performance scores. Moreover, we only considered models that provided forecasts for at least 50% of the target population. In Table 1 we present, for each model, the values of *rel_ae*, *rel_wis*, the 50% and 95% interval coverage and the number of targets in common between EpiLearn and the model used to compute the indicators. In Fig 5 we show the actual observed weekly disease incidence by country during the time span evaluated in the comparative results from the European COVID-19 Forecast Hub, this period showed challenging behaviors in disease incidence trends. The baseline value of rel_ae and rel_wis equal to 1 is the constant prediction, which is expected to be beaten by all more sophisticated methods. The lower the values of these indicators, the better the model performance. The result of the ensemble model is usually considered as the best option as argued in [20]. In the Hub, the proposed *EpiLearn* technique corresponds to the named AMM-EpiInvert team. In Table 1 we observe that, except for the horizon of 4 weeks, *EpiLearn* obtains the best results for *rel_ae* and *rel_wis*, specially for the horizons of 1 and 2 weeks. The value of the 50% confidence interval coverage obtained by *EpiLearn* is not so good, but this indicator does not take into account how far the estimation stands from the confidence interval. The low value of *rel_wis* indicates that, globally, the estimation obtained by *EpiLearn* are quite close to the confidence intervals. In Fig 6 we plot the values of *rel_ae* and *rel_wis* presented in Table 1.

The EpiInvert method performs a decomposition of the past incidence curve into a trend and a noise component, after correction of the weekly bias. In the above error estimations, we only used the trend curves, because what is forecast also is a trend. We did not compute the additional error between prediction and ground truth that is caused by the noise component. Our above error prediction therefore only addresses the method's bias, namely the observed variability of the future trends following a given past trend interval. A further refinement of the method should take into account the noise residual computed by EpiInvert for a given incidence curve, estimate its model, and deduce a noise variance for the prediction. This noise variance should be added to the method bias variance.

A great advantage of using the scores published by the European Hub is that such scores cannot be manipulated. They represent a fair quality comparison framework for the models performance.

**Table 1. Comparative results.** In bold the best result of each column for each week for the main quality measurements promoted by the European hub: *rel_ae* and *rel_wis* (for both quality criteria, the lower the better).

| horizon | model | *rel_ae* | *rel_wis* | cov 50 | cov 95 | N. targets |
|---|---|---|---|---|---|---|
| 1 week | EuroCOVIDhub-baseline | 1 | 1 | 0.604 | 1.000 | 728 |
| 1 week | EuroCOVIDhub-ensemble | 0.818 | 0.701 | 0.492 | 0.940 | 705 |
| 1 week | AMM-EpiInvert (EpiLearn) | **0.744** | **0.647** | 0.328 | 0.890 | 728 |
| 1 week | ILM-EKF | 7.354 | 7.618 | 0.387 | 0.846 | 688 |
| 1 week | MUNI-ARIMA | 1.005 | 0.947 | 0.484 | 0.952 | 649 |
| 1 week | USC-SIkJalpha | 1.856 | 2.002 | 0.162 | 0.478 | 696 |
| 1 week | epiforecasts-weeklygrowth | 4.839 | 4.342 | 0.747 | 0.964 | 501 |
| 1 week | SDSC_ISG-TrendModel | 0.984 | 1.033 | 0.602 | 0.883 | 703 |
| 2 weeks | EuroCOVIDhub-baseline | 1 | 1 | 0.730 | 0.999 | 697 |
| 2 weeks | EuroCOVIDhub-ensemble | 0.831 | 0.768 | 0.490 | 0.921 | 668 |
| 2 weeks | AMM-EpiInvert (EpiLearn) | **0.795** | **0.717** | 0.370 | 0.899 | 697 |
| 2 weeks | ILM-EKF | 8.374 | 10.817 | 0.355 | 0.803 | 656 |
| 2 weeks | MUNI-ARIMA | 1.020 | 0.988 | 0.557 | 0.960 | 618 |
| 2 weeks | USC-SIkJalpha | 2.655 | 2.882 | 0.123 | 0.374 | 660 |
| 2 weeks | epiforecasts-weeklygrowth | 3.764 | 4.291 | 0.712 | 0.973 | 479 |
| 3 weeks | EuroCOVIDhub-baseline | 1 | 1 | 0.751 | 0.996 | 668 |
| 3 weeks | EuroCOVIDhub-ensemble | 0.988 | 0.950 | 0.478 | 0.911 | 640 |
| 3 weeks | AMM-EpiInvert (EpiLearn) | **0.937** | **0.889** | 0.385 | 0.896 | 668 |
| 3 weeks | ILM-EKF | 8.860 | 12.046 | 0.322 | 0.763 | 628 |
| 3 weeks | MUNI-ARIMA | 1.001 | 0.943 | 0.581 | 0.950 | 585 |
| 3 weeks | USC-SIkJalpha | 5.339 | 5.158 | 0.117 | 0.340 | 633 |
| 3 weeks | epiforecasts-weeklygrowth | 2.209 | 2.103 | 0.676 | 0.974 | 457 |
| 4 weeks | EuroCOVIDhub-baseline | 1 | 1 | 0.777 | 0.994 | 633 |
| 4 weeks | EuroCOVIDhub-ensemble | 1.105 | 1.106 | 0.482 | 0.911 | 604 |
| 4 weeks | AMM-EpiInvert (EpiLearn) | 1.111 | 1.081 | 0.387 | 0.903 | 633 |
| 4 weeks | ILM-EKF | 8.618 | 12.049 | 0.317 | 0.755 | 596 |
| 4 weeks | MUNI-ARIMA | **0.952** | **0.912** | 0.565 | 0.964 | 549 |
| 4 weeks | USC-SIkJalpha | 4.684 | 5.185 | 0.107 | 0.323 | 598 |
| 4 weeks | epiforecasts-weeklygrowth | 2.059 | 2.613 | 0.685 | 0.980 | 454 |

## Discussion

In this section we review and discuss the properties and assumptions of the most relevant forecasting methods, and link them when possible to methods and results published weekly in the European Covid-19 Forecast Hub [17].

## ARIMA

The ARMA (AutoRegressive Moving Average) and ARIMA (AutoRegressive Integrated Moving Average) models are the backbone of many forecasting methods and are implemented through the popular R package [22]. In the European Hub Forecast initiative, the *epiforecasts-weeklygrowth* team [23] uses a Bayesian ARIMA model on weekly incidence data. ARIMA is arguably the most popular forecasting model for COVID-19, and has been applied with a country-specific optimization of parameters. For example the MUNI-ARIMA [24] team participating to the European Hub Forecast initiative uses an "ARIMA model with outlier

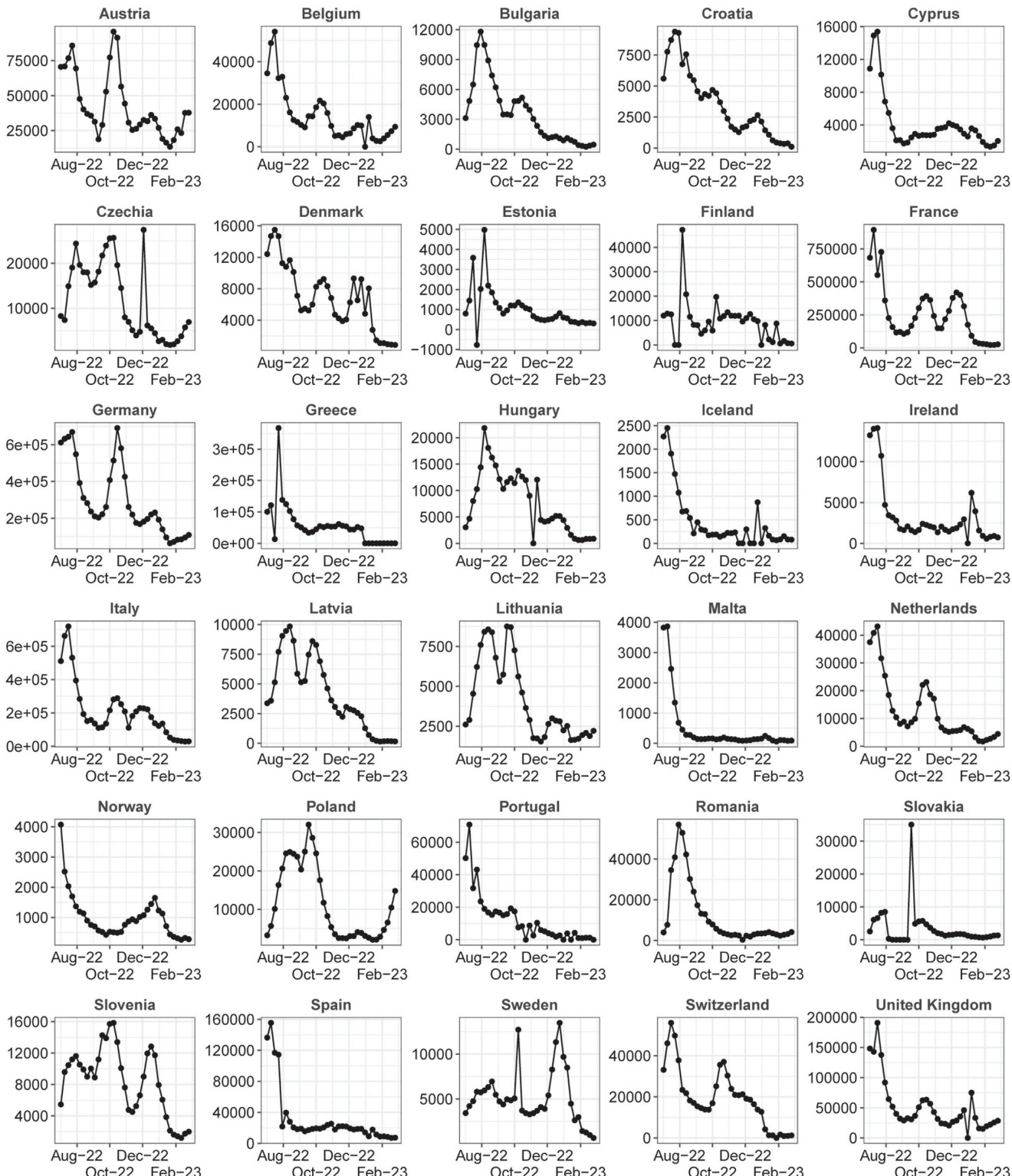

**Fig 5. Figure showing the actual observed weekly disease incidence by country during the time span evaluated in the comparative results from the European COVID-19 Forecast Hub.** This period showed challenging behaviors in disease incidence trends.

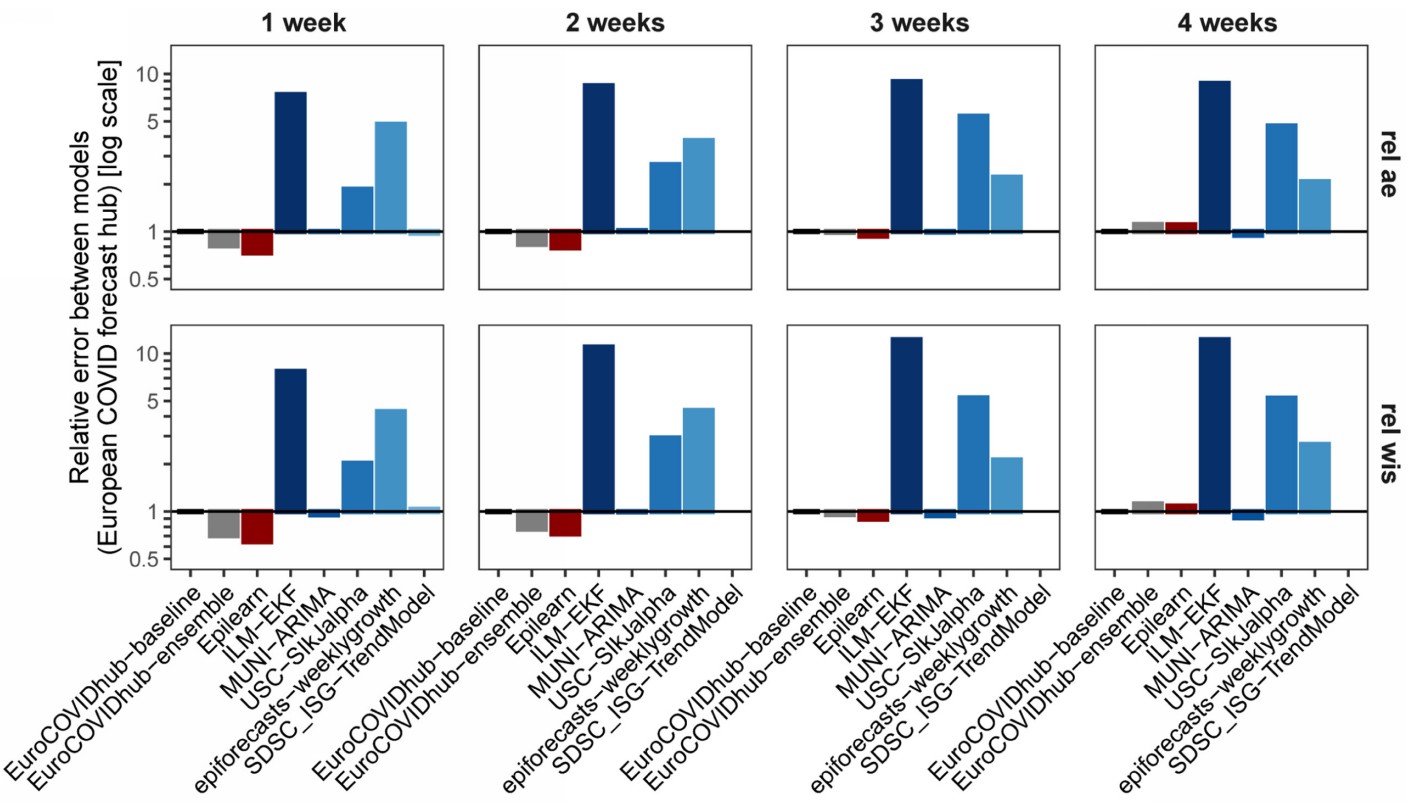

**Fig 6. Plot of the relative absolute error (*rel_ae*) and the relative weighted interval score (*rel_wis*) presented in Table 1 using the evaluation data provided by the COVID-19 European Hub.**

detection fitted to transformed weekly aggregated series". This method is one of the best performing methods as illustrated in Fig 6.

An extension of ARIMA, SARIMA (seasonal ARIMA) is a combination of two ARIMA models. This method was tested for forecasting the global COVID-19 incidence in [25, 26]. It has been used and compared to ARIMA for COVID-19 forecast in [27].

## Compartmental epidemiological models (SIR, SEIR, SIRD, SEIARD and SUIHTER)

Compartmental models are *in silico* simulation models that consider the population as a collection of compartments, for example in the case of SEIARD : *S* susceptible, *E* exposed, *I* infected, *A* asymptomatic, *R* recovered and *D* dead. Initially designed for epidemic modeling, the SIR model and its variants have since been adapted to forecasting the future evolution of the pandemic from an estimated starting point. The model's parameters are estimated from the past incidence, and the model is then applied forward to simulate the future. This method has been developed for SIR [28, 29], SEIR [30, 31], SIRD [32], SEIARD [33] and SUIHTER [34, 35].

## Regression models

**The Richards model** [36] is a 2-parameter simple logistic growth model including a scaling parameter. This model is used in [37] as a parametric regression model for the modeling of incidence indicators. The incidence distribution is modeled by an appropriate Poisson or

Negative Binomial. It is also used in [38] for estimating the regional propagation of COVID-19 in Italy and in [39] for recurrent forecasting in Europe. **The Gompertz model** was originally proposed to explain human mortality curves and has been further employed in the description of growth processes. Modeling the cumulative cases of Covid-19, it is used for COVID-19 forecast in [40] and [41]. This model was implemented in the European Hub Forecast initiative through the BIOCOMSC-Gompertz method [40]. **The composite logistic growth model (CLM)** [42] is another regression model, a variant of which is used by the RobertWalraven-ESG [43] team participating to the European Hub Forecast initiative. Its results are illustrated in Fig 6. **The sub-epidemic model** is the most flexible extension of the previous models used for forecasting [42] This sub-epidemic wave model supports complex epidemic trajectories shaped by multiple underlying sub-epidemics modeled by the GLM.

## Short term prediction by the renewal equation, linear extrapolation

The approach proposed in [44] to forecasting future COVID-19 cases involves 1) modeling the incidence using a Poisson distribution for the daily incidence number, and a gamma distribution for the series interval; 2) estimating the effective reproduction number assuming its value stays constant during a short time interval (by the EpiEstim method [7]); and 3) using the renewal equation, drawing future incidence cases from their posterior distributions, assuming that the current transmission rate will stay the same, or change by a certain degree. A similar forecast method is involved in [45] which compares human and machine forecasts in Germany and Poland. The authors use a Bayesian model from the EpiNow2 R package (version 1.3.3) to predict reported cases. Epinow [10] estimates the effective reproduction number $R_t$. The future infections are computed by the Fraser renewal equation as a weighted sum of past infection multiplied by $R_t$. In the comparison, $R_t$ is assumed to stay constant beyond the forecast date. The conclusion of this paper is that an average of human experts' forecasts performs better. Similarly, the USC-SIkJalpha [16] and ILM-EKF [46] teams participating to the European Hub Forecast initiative use the renewal equation (the second mentioned group also involves a Kalman filter in its prediction). Its results are illustrated in Fig 6. Lastly, the SDSC_ISG-TrendModel [47] team, also participating to the European Hub Forecast initiative, is a trend extrapolation which starts by decomposing the incidence curve into three components: the trend, a seasonal component and noise. Then the model predicts daily cases using linear extrapolation on the linear or log scale of the underlying trend estimated by a robust LOESS seasonal-trend decomposition model. Its results are illustrated in Fig 6, where only the results for the first two weeks are available.

## Aggregation of estimators (ensemble methods)

The idea of agregation methods, sometimes also called *ensemble* methods is to build a prediction model by combining the strengths of a collection of simpler base models called weak learners [48]. In [49] the use of ensemble models was evaluated for influenza seasons and it was concluded that the ensemble methods average performance is similar to the best of the component models, but offers more consistent performance across seasons than the component models. The European Covid-19 Forecast Hub [17] also proposes "an ensemble, or model average, of submitted forecasts to the European COVID-19 Forecast Hub", described in [20]. In it, the teams submit weekly forecasts for COVID-19 cases and deaths in up to 32 countries for the next week and the three following weeks. The teams also submit standardized quantiles of their predictive distribution. In the *ensemble forecast*, each predictive quantile is calculated as the equally-weighted median of all individual models' predictive quantiles. The performance of each model is evaluated with the relative Weighted Interval Score (WIS),

comparing a models' forecast accuracy relative to all other models (see section for the formula of WIS). In [20], the authors report that the ensemble performed better on relative WIS than 84% of participating models' forecasts of incident cases (with a total N = 862), and 92% of participating models' forecasts of deaths (N = 746). In view of this, we shall pay a special attention to the comparison of the model proposed here with the ensemble model, as illustrated on Fig 6.

## Global learning

The idea of Global learning is to predict jointly an ensemble of time series with similar characteristics [50]. Each time series is time-delay embedded and stacked together before fitting a single linear autoregressive model. The dimension of the embedding is tuned by temporal validation. The same method is used in [51], which proposes to estimate a time lag between two countries after finding an optimal dynamic time warping between their incidence curves. This procedure allows an elastic adjustment of the time axis to find similar but phase-shifted sequences. Then the incidence curve of the leading country is used to extend toward future the incidence curve of the other. This group of methods can be seen as a direct antecedent of the method proposed here. Indeed, our method (implicitly) estimates time lags between past incidence curves of different countries and the one that we want to extend before exploiting the "future" samples of these time shifted incidence curves to predict the future of our target incidence.

## Conclusion

Given the large number of factors that can influence a future evolution, forecasting the evolution of the incidence curve is clearly difficult. We saw in the discussion section that most standard approaches estimate the parameters of an evolution model (ARIMA, SIR, a logistic curve). In this work, we proposed *EpiLearn*, a method following a more empirical approach that estimates the forecast by a learning procedure using many samples of past incidences evolution in many countries. Using *EpiInvert*, an incidence decomposition method, we removed first the strong administrative weekly bias from the original raw incidence to estimate a smooth incidence trend curve. Using a large database of incidence trends, the forecast is computed as the median of the closest curves, in the past, to the current incidence trend curve. We observed that the size of the estimated empiric confidence interval grows quickly with the number of forecast days. For a 28 days forecast the size of the confidence interval becomes very large, and this is confirmed weekly by our results in the European hub [17]. These results place *EpiInvert* among the very best methods in the period and regions analyzed. We observed that the prediction of all methods may miss the forecast target by a large margin in the three and four weeks horizon. Nevertheless, they seem to be reliable and useful to predict the pandemic in a two-week horizon.

The proposed method might be improved in several ways by taking into account additional relevant factors before comparing time sequences. Indeed, our obtained confidence intervals were based on a global distribution of relative errors. However, the size of relative errors might vary depending on the trend and magnitude of the query curve. It might be interesting to explore this by adding a distance of the average incidences as an additional term in the distance between incidences given in equation (6). In addition, the forecasting could benefit from additional knowledge about implementation or changes of the social distancing policy and evolution of the virus' contagiousness (as was observed with the emergence of Omicron). Digging into these aspects requires a far ranging overhaul of our experimental protocol which belongs to our future plans.

## Author Contributions

**Conceptualization:** Jean-David Morel, Jean-Michel Morel, Luis Alvarez.

**Data curation:** Jean-David Morel, Jean-Michel Morel, Luis Alvarez.

**Formal analysis:** Jean-David Morel, Jean-Michel Morel, Luis Alvarez.

**Funding acquisition:** Jean-David Morel, Jean-Michel Morel, Luis Alvarez.

**Investigation:** Jean-David Morel, Jean-Michel Morel, Luis Alvarez.

**Methodology:** Jean-David Morel, Jean-Michel Morel, Luis Alvarez.

**Project administration:** Jean-David Morel, Jean-Michel Morel, Luis Alvarez.

**Resources:** Jean-David Morel, Jean-Michel Morel, Luis Alvarez.

**Software:** Jean-David Morel, Jean-Michel Morel, Luis Alvarez.

**Supervision:** Jean-David Morel, Jean-Michel Morel, Luis Alvarez.

**Validation:** Jean-David Morel, Jean-Michel Morel, Luis Alvarez.

**Visualization:** Jean-David Morel, Jean-Michel Morel, Luis Alvarez.

**Writing – original draft:** Jean-David Morel, Jean-Michel Morel, Luis Alvarez.

**Writing – review & editing:** Jean-David Morel, Jean-Michel Morel, Luis Alvarez.

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
