## [Decision Letter · Decision Letter 0]

8 Feb 2023

Dear Prof. Morel,

Thank you very much for submitting your manuscript "Learning from the past: a short term forecast method for the COVID-19 incidence curve" for consideration at PLOS Computational Biology.

As with all papers reviewed by the journal, your manuscript was reviewed by members of the editorial board and by several independent reviewers. In light of the reviews (below this email), we would like to invite the resubmission of a significantly-revised version that takes into account the reviewers' comments.

Please pay special attention to Reviewer 1's comment about overall clarity and Reviewer 2's comments about review of related literature and the period of forecasting evaluation. These issues will need to be addressed compellingly for a positive outcome to be possible.

We cannot make any decision about publication until we have seen the revised manuscript and your response to the reviewers' comments. Your revised manuscript is also likely to be sent to reviewers for further evaluation.

Sincerely,

Alex Perkins

Academic Editor

PLOS Computational Biology

Lucy Houghton

Staff

PLOS Computational Biology

Reviewer's Responses to Questions

**Comments to the Authors:**

Reviewer #1: This article deals with the covid-19 data for several countries (France, Germany, USA UK ). Unfortunately, this article is not clearly written, so I can not give an opinion at this level.

From the abstract, their goal is to use some method called EpiInvertForecast, which is not explained in the paper.

After (briefly) checking their explanation on the webpage dedicated to EpiInvertForecast, it looks like a series of standard statistical technics (I do not see anything new (possibly I miss understood it).

In the "result" section, I do not understand where their results are.

At the moment, the paper is too confusing to say anything.

Therefore I recommend significant modifications to clarify it.

Reviewer #2: Summary:

This article presents a method for forecasting infectious disease using a variation on the “method of analogues”, and applies that modeling approach to forecasting COVID-19 incidence in Europe. In my view, this could eventually be publishable work that applies a variation on an existing method to a novel setting. However, several points outlined in more detail below would need to be addressed before publication. Most importantly: (1) the literature review omits closely related work; (2) the evaluation time frame is far too short; and (3) a large section of the discussion could be omitted or much abbreviated. I don’t see a satisfactory way to address the second of these points without either (a) waiting some time to gather more data on model performance, or (b) expanding the application to additional time frames and/or other locations [while taking care to make proper use of the versions of data that would have been available in real time].

Major comments:

- The introduction and literature review do not include some relevant literature. The method that is introduced in this paper is a variation on the “method of analogues”, which has been applied to infectious disease forecasting in the past for both point forecasts and probabilistic forecasts; see the references in the points below. The authors should properly review the literature in the introduction and situate their methodological contributions appropriately. (To be clear: there are methodological innovations in the present work; I would just like to see the connection made to past similar work.)

(1) Viboud, Cécile, et al. "Prediction of the spread of influenza epidemics by the method of analogues." American Journal of Epidemiology 158.10 (2003): 996-1006.

(2) Ray, Evan L., et al. "Infectious disease prediction with kernel conditional density estimation." Statistics in medicine 36.30 (2017): 4908-4929.

- The evaluation time period of 2 months is extremely short. I would argue that this time frame is far too short to obtain reliable results about model performance. Evaluations of forecasts in similar settings have found that models can appear to do well for spans of several months at a time and then have substantial errors.

- The level of detail in the discussion section seems out of place. It is valuable to include references to this previous work, but I would argue that this literature review could be shortened to three paragraphs or less with no impact on the communication of the main body of the work. For example, Table 1 could be omitted entirely; the parameter values that were selected for ARIMA models in 14 other published papers have no bearing on the reader’s understanding or interpretation of the results for the methods described in this manuscript. Similarly, the commentary on compartmental models is not relevant or helpful to the main point of the current paper (and I’m not convinced by the authors’ arguments, as, e.g., depletion of susceptibles is a major driving force in the forecasts that are generated by these models, and is typically estimated by fitting the model to a time span of many weeks).

Additional substantive comments

- On page 4, the authors write, “To add a curve of this type to the database, two conditions were imposed: the first was that the minimum time interval of the resulting sequence to apply EpiInvert was 150 days. The second condition was that the mean of the 56 values of the sequence must be larger than 1000. (Small averages can correspond to nonthreatening or neglected stages of the epidemic…)” I have a question about each of these conditions:

(1) I don’t understand the statement of the first condition; can the authors please clarify what is meant by “the minimum time interval of the resulting sequence to apply EpiInvert”?

(2) It would seem that eliminating reference curves with low incidence from the database would systematically bias the forecasts at times with low incidence. Can the authors comment on this?

- In Eq (4), it appears that the forecasts are scaled by the factor s_{28}/i^k_{28}, but in Equations (1) and (2) normalization was done so that the average of the first 28 days summed to 1. Can the authors comment on why two different scaling procedures were used? Would it not be better to pick one to use more consistently?

- Although forecasts are produced at horizons of 1 through 28 days ahead, the selection of the tuning parameters N_median and mu was done by examining only the errors at horizons of 1 through 14 days ahead. Why? Would different tuning parameters have been selected if evaluation was done based on the full forecast horizon that was used?

- How well calibrated were the forecasts? Could the authors include, e.g., an evaluation of interval coverage rates or one-sided quantile coverage rates?

- The method accounts for a trend (and uncertainty about the trend) and a weekly cycle in reporting. However, it seems that additional noise is not captured. As described in the discussion section, the EpiInvert method that is used for preprocessing decomposes the series into trend, seasonality, and noise components; the noise seems to be lost in subsequent processing in the proposed algorithm, though.

- “Since EpiLearn forecasts the daily incidence, the weekly forecast is obtained by summing the forecasted raw daily incidence given by (5). The quantiles of the associated weekly distributions are computed on the registered database of incidence curves by extending the procedure of section which computes the confidence intervals of the forecasted incidence curve.”

(1) The procedure for obtaining weekly quantiles is not completely clear: do you aggregate to the weekly scale first and then compute quantiles, or do you compute quantiles on the daily scale and then aggregate? I think you are doing the second of these, but it would be helpful to be precise.

(2) If indeed you are calculating quantiles on a daily scale and then aggregating, I think that you are implicitly assuming perfect dependence in the forecast distributions across days. This assumption is not true, but this results in an inflation of uncertainty. Please comment on this.

- It would be helpful to include a figure showing the actual observed disease incidence during the time span evaluated in the section on Comparative results from the European COVID-19 Forecast Hub. Was this a time with “interesting” behavior in disease incidence trends?

- As the authors discuss in the Materials and Methods section, infectious disease surveillance data are subject to revision. In evaluations of forecasting methods, it is critical to address this by using the version of the data that would have been available in real time when producing forecasts. It is my understanding that this was done, but a clear statement of this would be beneficial. (If it was not done, the analysis should be reworked.)

Minor comments:

- Around Eq (5), it would be helpful to define % as the modulus operator.

- The sentence “Fig 3 shows the function f (x) = e−0.0475x which determines the weight assigned to each day in the past in the distance estimation.” on page 7 would better belong just after Eq (3), where the exponential weighting is first described.

- There is some notational inconsistency in the use of the subscript m:

(1) In Eq (6), the authors define \\hat{u}_m to denote a forecast at horizon m, but in Eq (7) and (9) a subscript of m+28 is used.

(2) Later, m is used yet again to index the forecast model

- Typo near the top of page 8 where the subscript 0.0975 is used rather than 0.975

- Throughout the manuscript, section references are missing

- There are minor spelling and grammatical issues throughout the manuscript that would benefit from the attention of an editor.

**Have the authors made all data and (if applicable) computational code underlying the findings in their manuscript fully available?**

Reviewer #1: Yes

Reviewer #2: Yes

PLOS authors have the option to publish the peer review history of their article (what does this mean?). If published, this will include your full peer review and any attached files.

Reviewer #1: No

Reviewer #2: **Yes: **Evan L. Ray
---

## [Decision Letter · Decision Letter 1]

14 Apr 2023

Dear Prof. Morel,

Thank you very much for submitting your manuscript "Learning from the past: a short term forecast method for the COVID-19 incidence curve" for consideration at PLOS Computational Biology.

As with all papers reviewed by the journal, your manuscript was reviewed by members of the editorial board and by several independent reviewers. In light of the reviews (below this email), we would like to invite the resubmission of a significantly-revised version that takes into account the reviewers' comments.

We cannot make any decision about publication until we have seen the revised manuscript and your response to the reviewers' comments. Your revised manuscript is also likely to be sent to reviewers for further evaluation.

Sincerely,

Alex Perkins

Academic Editor

PLOS Computational Biology

Lucy Houghton

Staff

PLOS Computational Biology

Reviewer's Responses to Questions

**Comments to the Authors:**

Reviewer #2: Summary:

I continue to believe that the work described in this manuscript should be published; it describes a successful application of a method with demonstrated strong performance relative to other methods for forecasting COVID-19 incidence. The majority of my comments on the first submission have been successfully addressed in this revision, but I still have a number of relatively minor comments remaining, which primarily relate to the readability and clarity of the paper rather than substantive comments on the methods or evaluation. That said, I am suggesting another round of “major revisions” due to the number of comments here.

Comments on methods:

1. Although nearly all of the points I raised in the initial submission were addressed, there is one I would like to follow up on. I reproduce the discussion here.

My initial comment was “The method accounts for a trend (and uncertainty about the trend) and a weekly cycle in reporting. However, it seems that additional noise is not captured. As described in the discussion section, the EpiInvert method that is used for preprocessing decomposes the series into trend, seasonality, and noise components; the noise seems to be lost in subsequent processing in the proposed algorithm, though.”

The authors’ response was: “the noise model for the trend curve estimated by EpiInvert does not include any forecast procedure. In this paper, we do not formulate any model for the noise of the forecast of the trend curve (that could be far from trivial). We just study, empirically, the forecast confidence intervals.”

My follow-up comment: At minimum, it seems worth explicitly acknowledging the fact that noise around the trend is not modeled as a limitation of the method. It seems possible that this might be a contributing factor to the lower-than-nominal interval coverage rates reported in the updated Table 1. It would also be nice to include any ideas you have for possible approaches to addressing this in the discussion for future work. For example, might it be possible to extract residuals and model their distribution on a per-location basis?

2. In the paragraph describing the Data preprocessing method, there is a comment that the serial interval is observable. What is meant by this? It is my impression that most public health surveillance data sets do not have sufficient resolution to observe the serial interval, since we do not have information about secondary cases. Are you simply using estimates of the serial interval reported in the cited reference, Ma et al.? If so, it would be helpful to state this and the specific value used.

3. Confidence intervals are currently based on a global distribution of relative errors. However, I might imagine that the size of relative errors might vary depending on the epidemic phase. It might be interesting to examine this with a plot similar to Fig 4 examining the distribution of relative errors as obtained for example in phases of steady low disease incidence, increasing incidence, high incidence, and low incidence. The motivating question here is whether it might make sense to calculate prediction intervals based on quantiles of the selected reference trajectories from the database rather than global relative errors?

Comments on organization:

4. I suggest moving the contents of the “discussion” section up to the introduction. This material is really a literature review, and so I think it would be more standard to place it near the beginning of the article.

5. I suggest that the first several subsections of the results would be better placed under Materials and Methods, including “Incidence trend curves database construction using EpiInvert”, “Normalization of the database incidence curves”, “Computing the distance between curves”, “Forecasting using a median of the closest database curves”, “Choice of the method parameters”, and “Empirical confidence intervals.” The sections describe aspects of the methodology rather than results.

Comments on notation:

6. In the section on “Choice of method parameters”, it seems like there is some inconsistency in notation around equations (9) and (10). The second sentence reads “Set, for each i^k in the database, …”, but i^k does not appear in Eq (9). Do you mean for each s^k in the database? In the following sentence you have s^k without a hat, and Eq (10) refers to i^k_{d+28} which I think should correspond to \\hat{s}^k_{d+28} in the notation of Eq (9) or s^k in the text just after Eq (9). I have similar questions about the notation in the following section on “Empirical confidence intervals”: is s^k being used here to refer to the first 28 entries of i^k? Could we then just pick one letter, either using i^k with appropriate indexes or s^k and s^f? Am noting that the lack of a k index on s^f is also confusing. Summing up: I think it would be helpful to establish early on a consistent notation for a curve that is being forecasted and a reference curve from the database, and use this standardized notation throughout.

7. Similarly, it seems like it might be possible to establish a single notation for a relative forecast error to use throughout the sections on choice of method parameter and empirical confidence intervals. We currently have e^d,k and e_d,k which differ only in whether the numerator involves an absolute value and whether the error is regarded as a function of the model hyperparameters N_{median} and mu. Could this discussion be streamlined in terms of notation, for example using only the e_d,k notation that is currently introduced in Eq (12) currently, and in Eq (11) summing the absolute errors and discussing the dependence of the hyperparameters in the text?

8. It is not clear why in Eq (9) the numerator involves \\hat{s}^k_28, in contrast to the numerator of s_28 (without a hat) that is used when generating forecasts in Eq (7). My first thought was that this was done to remove scale effects in the unnormalized s, but that should be unnecessary since the relative error metric in Eq (10) should account for this already. Could we then simplify the presentation and notation by re-using the notation of Eq (7) for a forecast rather than introducing the new notation \\hat{u}^k_d?

9. In Eq (6), when computing the similarity of the normalized vectors \\hat{s} and i^k, should the comparison be based on i^k_j, i.e., we compare entry j of \\hat{s} to entry j of i^k? The subscript p seems out of place.

Misc. minor comments on word choice, etc:

10. In introduction: “the methods that were publicly available…” -> capitalize “The”

11. “Our learning technique is widely different from previous methods introduced in the literature.” The revised manuscript now includes a review of a few similar papers, including one paper that was applied to forecasting COVID-19. Maybe revise the statement here to something that is a little more qualified, e.g. “Our learning technique is different in structure from most previous methods introduced in the literature.”

12. ”A weekly and festive bias corrected incidence …” Unclear what is meant by “festive” here; suggest another word choice.

13. In the section on “Forecasting using a median of the closest database curves”, change “criterium” to “criterion”.

14. There is text that did not render properly at the bottom of the section on “Choice of method parameters”

15. Fig 6 is somewhat confusing because the bars for relative WIS are not shown for the ILM-EKF method; this makes it appear that the method is quite good, but I think that actually its performance is “off the charts” in terms of poor performance and the bar was not displayed.

16. In the discussion, ensemble methods are described as follows: “At every step, the ensemble fits a new learner to the difference between the observed response and the aggregated prediction of all learners grown previously.” This appears to be a description of boosting methods, which is one of many approaches to ensembling, and is not the method used in references 17 and 20. I suggest removing this sentence, as well as the sentence immediately after it.

17. I appreciate the addition of the new Figure 5; this provides valuable context. The font size is quite small, though. It might be helpful to make the figure slightly larger or show data only for selected locations (perhaps with all locations shown in a supplemental figure), to allow for a larger font size.

**Have the authors made all data and (if applicable) computational code underlying the findings in their manuscript fully available?**

Reviewer #2: Yes

PLOS authors have the option to publish the peer review history of their article (what does this mean?). If published, this will include your full peer review and any attached files.

Reviewer #2: **Yes: **Evan L. Ray
---

## [Editor Report · Decision Letter 2]

22 May 2023

Dear Prof. Morel,

We are pleased to inform you that your manuscript 'Learning from the past: a short term forecast method for the COVID-19 incidence curve' has been provisionally accepted for publication in PLOS Computational Biology.

Best regards,

Alex Perkins

Academic Editor

PLOS Computational Biology

Lucy Houghton

Staff

PLOS Computational Biology

---

## [Editor Report · Acceptance letter]

19 Jun 2023

PCOMPBIOL-D-22-01772R2 

Learning from the past: a short term forecast method for the COVID-19 incidence curve

Dear Dr Morel,

I am pleased to inform you that your manuscript has been formally accepted for publication in PLOS Computational Biology. Your manuscript is now with our production department and you will be notified of the publication date in due course.

With kind regards,

Timea Kemeri-Szekernyes
